# Deformation Parameters of the Heart in Endurance Athletes and in Patients with Dilated Cardiomyopathy—A Cardiac Magnetic Resonance Study

**DOI:** 10.3390/diagnostics11020374

**Published:** 2021-02-22

**Authors:** Łukasz A. Małek, Łukasz Mazurkiewicz, Mikołaj Marszałek, Marzena Barczuk-Falęcka, Jenny E. Simon, Jacek Grzybowski, Barbara Miłosz-Wieczorek, Marek Postuła, Magdalena Marczak

**Affiliations:** 1Department of Epidemiology Cardiovascular Disease Prevention and Health Promotion, National Institute of Cardiology, 04-635 Warsaw, Poland; 2Department of Cardiomyopathy, National Institute of Cardiology, 04-628 Warsaw, Poland; lmazurkiewicz@ikard.pl (Ł.M.); jgrzybowski@ikard.pl (J.G.); 3Medical University of Warsaw, 02-091 Warsaw, Poland; s075803@student.wum.edu.pl (M.M.); s079363@student.wum.edu.pl (J.E.S.); 4Department of Pediatric Radiology, Medical University of Warsaw, 02-091 Warsaw, Poland; marz.barczuk@gmail.com; 5Department of Radiology, National Institute of Cardiology, 04-628 Warsaw, Poland; barbara-milosz@tlen.pl (B.M.-W.); mmarczak@ikard.pl (M.M.); 6Department of Experimental and Clinical Pharmacology Centre for Preclinical Research and Technology (CEPT), Medical University of Warsaw, 02-097 Warsaw, Poland; mpostula@wum.edu.pl

**Keywords:** athlete’s heart, dilated cardiomyopathy, feature tracking, cardiac magnetic resonance

## Abstract

A better understanding of the left ventricle (LV) and right ventricle (RV) functioning would help with the differentiation between athlete’s heart and dilated cardiomyopathy (DCM). We aimed to analyse deformation parameters in endurance athletes relative to patients with DCM using cardiac magnetic resonance feature tracking (CMR-FT). The study included males of a similar age: 22 ultramarathon runners, 22 patients with DCM and 21 sedentary healthy controls (41 ± 9 years). The analysed parameters were peak LV global longitudinal, circumferential and radial strains (GLS, GCS and GRS, respectively); peak LV torsion; peak RV GLS. The peak LV GLS was similar in controls and athletes, but lower in DCM (*p* < 0.0001). Peak LV GCS and GRS decreased from controls to DCM (both *p* < 0.0001). The best value for differentiation between DCM and other groups was found for the LV ejection fraction (area under the curve (AUC) = 0.990, *p* = 0.0001, with 90.9% sensitivity and 100% specificity for ≤53%) and the peak LV GRS diastolic rate (AUC = 0.987, *p* = 0.0001, with 100% sensitivity and 88.4% specificity for >−1.27 s^−1^). The peak LV GRS diastolic rate was the only independent predictor of DCM (*p* = 0.003). Distinctive deformation patterns that were typical for each of the analysed groups existed and can help to differentiate between athlete’s heart, a nonathletic heart and a dilated cardiomyopathy.

## 1. Introduction

It has been established that the hearts of endurance athletes undergo characteristic physiological remodelling, which primarily involves enlargement of all heart chambers and is sometimes accompanied by mild left ventricular hypertrophy, which is referred to as athlete’s heart (AH) [1,2]. In effect, the size of the left ventricle (LV) appears to exceed the upper reference values in approximately 40–70% of male endurance athletes, as demonstrated in echocardiographic and cardiac magnetic resonance (CMR) studies [2,3]. The size of the left ventricle can therefore be similar to that in patients with dilated cardiomyopathy (DCM). In contrast, however, an indicative marker of pathology, namely, a low LV ejection fraction, is rare, as it infrequently drops below 55% in AH [4]. Nonetheless, in athletes with a markedly enlarged LV and borderline LV systolic function, distinguishing between physiological remodelling and pathology is difficult [5]. Such diagnostic differentiation is clinically important, as DCM is one of the definable causes of sudden cardiac death in athletes, accounting for up to 8% of all cases [6,7]. A structured diagnostic algorithm has been proposed to differentiate between adaptive and maladaptive cardiac changes. It considers the presence of additional features favouring AH, such as (a) symmetrical right ventricular enlargement, (b) an asymptomatic course, (c) a lack of pathological findings on the electrocardiogram or Holter monitoring, (d) the presence of a significant LV ejection fraction improvement during exercise and (e) a lack of late gadolinium enhancement (LGE) other than at junction points on CMR [8,9,10,11]. Nevertheless, the differential diagnosis of early, preclinical phase DCM cases remains challenging.

Another parameter, which could facilitate the differentiation of AH from DCM is a better understanding of the cardiac deformational mechanics in both groups [9,10]. These may be assessed using echocardiographic speckle tracking (STE) or CMR feature tracking (CMR-FT). These tools enable the precise tracking of the systolic and diastolic multidirectional deformations and rotation of the heart [12,13]. Although several echocardiographic studies and a single CMR-FT study were done comparing the cardiac deformational mechanics of athletes to those of controls, no study heretofore has employed this tool to differentiate between athlete’s heart and DCM [10,14,15,16,17,18].

On these grounds, we aimed to retrospectively analyse the rotational mechanics of the hearts of ultramarathon runners relative to both patients with DCM and nonathletic controls using CMR-FT. Our objective was to expand on what is known of myocardial contraction in these groups, as well as to identify parameters that bear a potential value in the discrimination between adaptive and maladaptive cardiac changes.

## 2. Materials and Methods

### 2.1. Study Groups

The study included a group of 22 male amateur endurance athletes and a comparable number of sex- and age-matched patients with DCM, as well as healthy sedentary controls. The athletes were all long-term ultramarathon runners with at least 7 years of documented training, running an average of 70 km per week with frequent starts in competitions, which often exceeded 100 km in distance. They were all asymptomatic and free of cardiovascular disease, as verified using an electrocardiogram (ECG), cardiopulmonary exercise testing and CMR, which is presented in detail elsewhere [3]. All patients with DCM had a clinically confirmed disease and were under typical treatment for DCM in a specialised tertiary centre, where they underwent CMR. In order to eliminate severe cases of the disease, only DCM patients with an LV ejection fraction >30% and in a functional New York Heart Association (NYHA) class I were included in the analysis. Controls were healthy male subjects who volunteered to participate in the study but were not engaged in exercise beyond sporadic, recreational forms of physical activity.

### 2.2. CMR Study and Analysis

The CMR imaging was performed using a Siemens Magnetom Skyra 3 T scanner in athletes and controls and with a Siemens Magnetom Avanto Fit 1.5 T scanner in patients with DCM (Siemens, Erlangen, Germany). The protocol included initial scout images, followed by cine steady-state free precession (SSFP) breath-hold sequences in two-, three- and four-chamber views. The short axis was identified using the two- and four-chamber images and included the ventricles from the mitral and tricuspid valvular plane to the apex. This was followed by the administration of 0.1 mmol/kg of a gadolinium contrast agent (gadobutrol in Gadovist^®^, Bayer Pharma AG, Berlin, Germany) flushed with 30 mL of isotonic saline. Late gadolinium enhancement (LGE) images in three long axes and a stack of short-axis imaging planes were obtained with a breath-hold segmented inversion recovery sequence performed 10 min after the contrast injection. The inversion time was adjusted to completely null the normal myocardium (typically between 250 and 350 ms).

Images were retrospectively analysed with the use of dedicated software (cvi42, v. 5.11, Circle Cardiovascular Imaging, Calgary, AB, Canada) [13]. An automatic contour detection on the two-, three- and four-chamber long-axis slices and all short-axis slices was used, along with discrete manual adjustments to properly delineate the endo- and epicardial contours of both ventricles. These were used to calculate the end-diastolic and end-systolic ventricular volumes, stroke volume and ejection fraction of the left ventricle and right ventricle (RV) and, in the case of the LV, the myocardial mass. The values were then indexed to the body surface area.

The same contours were used for the analysis using CMR-FT. The peak global circumferential strain and global radial strains (GCS and GRS, respectively), and the peak GCS/GRS systolic and diastolic strain rates were derived from the analysis of a stack of short-axis slices. These images were also used to calculate the LV peak torsion and torsion rate, which reflects the difference between the clockwise rotation of the basal segments and the counterclockwise rotation of the apex during systole. The peak global longitudinal strain (GLS) and peak GLS systolic and diastolic strain rates of both ventricles were obtained from the combined analysis of three long-axis image projections.

The presence and location of LGE was assessed visually and classified by location (junction point, interventricular septum, inferolateral segments, other) and type (nonischaemic, i.e., mid-wall/subepicardial, or ischaemic). The analysis was blinded to the group assignment.

### 2.3. Statistical Methods

All results for the categorical variables are presented as a number and a percentage. Continuous variables are expressed as a mean with standard deviation (SD) or a median with an interquartile range (IQR), depending on the normality of the distribution, as assessed using the chi-squared test. Either the chi-squared test or Fisher’s exact test was used for the comparison of categorical variables, when appropriate. Student’s *t*-test or the Mann–Whitney test for unpaired samples were applied to compare two groups and the one-way ANOVA test or the Kruskal–Wallis test were used to compare three groups depending on the normality of the distribution, respectively. The receiver operation curve (ROC) analysis was computed to assess the discriminating value of the analysed parameters between the DCM and athlete’s heart/controls. Parameters with the highest discriminating value were included in the multivariable logistic regression to find independent predictors of DCM. The inter-reader variability was assessed using intraclass correlation coefficients (ICCs). All tests were two-sided with the significance level set to *p* < 0.05. The statistical analyses were performed using MedCalc statistical software 10.0.2.0 (MedCalc, Mariakerke, Belgium).

## 3. Results

### 3.1. Baseline Characteristics

Subjects from all groups were of similar age. The baseline CMR characteristics were typical for each studied group (Table 1).

All control subjects had normal LV and RV sizes, systolic function and LV masses [19]. Only one of them (8%) had a small junction point fibrosis, but it was considered to be benign. In comparison to the controls, the athletes had higher LV and RV sizes, a comparable systolic function of both ventricles and higher LV masses. Four of them (28%) had small LGEs of nonischaemic aetiology, mostly in the junction points and in one case in the inferolateral wall (subepicardial, most likely after prior asymptomatic myocarditis). Patients with DCM exhibited an enlarged LV size similar to that in athletes, but with lower LV ejection fractions and masses (Figure 1). Their RV sizes were lower than in athletes and comparable to controls, but with lower RV ejection fractions. In 50% of them, there was a mid-wall LGE in the inferolateral wall or the interventricular septum.

### 3.2. Deformation Mechanics of the LV

Peak GLS of the LV was similar in athletes and in controls, but lower in patients with DCM (Table 2, Figure 2).

A similar pattern was observed for the peak systolic GLS rate, while the peak diastolic GLS rate was similar in the athletes and in patients with DCM and lower than in the controls (Figure 3).

The peak LV GCS and peak LV GRS were lower in athletes and in patients with DCM in comparison to the controls (Figure 2). A similar pattern was observed for the peak systolic GCS and GRS rates, while the peak diastolic GLS rate was similar in the athletes and controls and lower than in patients with DCM (Figure 3).

Examples of the main LV deformation parameters as 4D reconstructions in single subjects from each group are supplied in a Appendix A.

Differences in the peak torsion and torsion rate were observed only between the controls and patients with DCM.

### 3.3. Deformation Mechanics of the RV

The peak GLS of the RV was lower in athletes in comparison to the controls (*p* = 0.028), as was the peak GLS systolic rate (*p* = 0.035). Besides this, no significant differences were found between the studied groups in terms of the RV deformation mechanics (Table 2).

### 3.4. Discriminating Ability of the Analysed Parameters

The ROC analysis included the parameters that best differentiated between the adaptive changes (athlete’s heart) and maladaptation (DCM) in the univariate analysis. It revealed that five parameters had a very high area under the curve (AUC) (>0.90, range 0.918–0.990) that was characterised by high sensitivity (72.7% to 100%—best for the peak LV GRS diastolic rate) and specificity (88.4–100%—best for the LV ejection fraction), as demonstrated in Table 3. Those five parameters were included in the multivariable logistic regression model, which showed that the only independent predictor of DCM was the peak LV GRS diastolic rate (*p* = 0.003).

Of note, the left and right ventricular sizes and the RV ejection fractions had only moderate accuracy in the discrimination between physiology and pathology, with low sensitivity for LV size and RV ejection fraction (50% and 59.1%) and low specificity for RV size (41.9%), as demonstrated in Table 3.

### 3.5. Inter-Reader Variability

The inter-reader variability was analysed using 20 randomly peaked studies from the whole sample. The ICCs for the assessment of the peak LV GLS, GCS and GRS were 0.95, 0.96 and 0.98 respectively. The ICCs for the assessment of the peak torsion and RV GLS were 0.91 and 0.92.

## 4. Discussion

Our study was the first to analyse the differences in the deformational mechanics of both ventricles of the heart in endurance athletes, non-athletic controls and patients with DCM. Previous studies, which were performed mostly using echocardiography, compared athletes to controls [9,10,14,15,16]. Thus far, few studies have used CMR-FT as a tool to analyse the deformational mechanics of the hearts of athletes [17,18]. In fact, the gap in the literature produced by the lack of direct comparison between the two studied groups addressed in this study has been referenced in the position statement on the use of cardiovascular imaging in the evaluation of athlete’s heart [10].

Sports cardiologists are occasionally faced with the need to differentiate pronounced forms of athlete’s heart—including a marked enlargement of the LV with borderline LV ejection fraction—from DCM, which, in the preclinical phase, can present in a similar form [5,9,10]. Different algorithms have been proposed to better characterise the grey area between athlete’s heart and DCM patterns in imaging [9,10,11]. These include the presence of other features of athlete’s heart, such as enlargement of the RV and the atria. Nonischaemic forms of LGE are more likely to be present in patients with DCM, as seen in our study. However, athletes may also present with subepicardial or mid-wall scarring that is likely postmyocarditis, which can further blur the clinical picture [20]. Unfortunately, steady-state imaging is often unable to clearly differentiate between both entities. Therefore, other markers must be taken into account, such as (a) the presence of symptoms, (b) a pathological electrocardiographic pattern, (c) the presence of arrhythmias during Holter monitoring, (d) increased biomarker levels (especially N-terminal prohormone of brain natriuretic peptide - NT-proBNP) or (e) a lack of significant improvement in the LV ejection fraction during exercise [11,21].

A deeper analysis of LV and RV systolic and diastolic function, including deformation parameters and rotational mechanics, may benefit the differentiation process. CMR appears to be a favourable and reproducible tool for this purpose due to its good signal-to-noise ratio, which allows for clear delineation of the heart structures and endo- and epicardial contour detection with modern postprocessing tools, as demonstrated in our study. Therefore, unlike with echocardiography, a reliable analysis does not have to be limited to GLS and can include other parameters, such as GCS, GRS or torsion, as well as RV mechanics. CMR is also one of the imaging methods that is recommended by experts for routine use in grey zone cases of athlete’s heart, as the analysis of deformation parameters comes at no additional cost and can be done in the postprocessing stage [9,10]. However, CMR parameters cannot directly be compared to echocardiographic ones and need their own normal values [22]. There are also significant differences between the vendors of feature-tracking software, as shown in previous studies, which does not make them interchangeable [23,24]. A previous study demonstrated that the most widely used parameter, namely LV GLS, is lower in CMR-FT than in speckle tracking imaging, as in our study [25].

We have demonstrated that each of the analysed groups has its unique deformational characteristics for left and right ventricular cardiac muscle. It has been previously shown that elite athletes have improved LV systolic performance at rest, which is characterised by significant shortening of the systolic time duration in comparison with sedentary controls, in association with a significant increase in the LV emptying velocity [26]. We have shown that this may be related to decreased peak LV GCS and GRS (mainly due to lower peak GCS and GRS systolic rates), while peak GLS and peak torsion did not change significantly, maintaining a normal LV ejection fraction. This was accompanied by lower RV GLS. Our findings are supported by previous research with the use of CMR-FT in endurance athletes that also showed similar LV GLS, lower LV GCS/GRS and lower RV GLS in comparison to controls [17,18]. However, unlike in the study by Swoboda et al., we did not observe lower torsion in our athletic group [17].

Pathological remodelling of the heart in DCM is characterised by enlargement of the LV, which is accompanied by decreased LV systolic performance at rest. A lower LV ejection fraction in patients with DCM in comparison to AH is an effect of further decreases in peak GCS and GRS, but also of a decrease in other parameters of systolic LV performance, such as peak LV GLS and peak torsion. In our study, peak LV GCS and GRS diastolic rates seemed to best discriminate physiological from pathological LV remodelling, with peak LV GRS diastolic rate being the only independent predictor of DCM. The peak LV GRS diastolic rate and LVEF can potentially be considered as complementary discriminatory parameters in the differentiation between DCM and AH, as the first one had perfect sensitivity and the other one showed perfect specificity, as demonstrated in our study.

From a pathophysiological standpoint, our results can be explained by the type of scarring present in patients with DCM, which is often diffuse and not visible with simple LGE imaging. GCS represents mid-wall fibers and GRS represents epicardial fibers, which are typically affected by the fibrosis observed in DCM, in contrast to GLS, which is more representative of subendocardial scarring that is typical for ischaemic cardiomyopathy [27,28].

Our study has some limitations. We included endurance athletes with enlarged heart chambers and nonathletic controls, but not athletes lacking features of AH as a comparison group [11]. Furthermore, there was a clear difference in the LV ejection fraction between the athletes and patients with DCM without many grey zone cases in terms of borderline ejection fractions [11]. Nevertheless, we believe that the finding of the LV GRS diastolic rate as the sole independent parameter predicting DCM, despite marked differences in the LV ejection fraction between the athletes/controls and patients with DCM, shows that deformation parameters can bear a significant role in a differential diagnosis between adaptive and maladaptive cardiac changes.

## 5. Conclusions

There is a distinctive deformation pattern that is typical of each of the analysed groups, which could help in differentiating between athlete’s heart, a nonathletic heart and dilated cardiomyopathy. The peak left ventricular global circumferential and radial strains, as well as their diastolic strain rates, are parameters that are characterised by very good sensitivity and specificity when differentiating between physiology and pathology. The peak left ventricular global radial diastolic strain rate was the only independent predictor of dilated cardiomyopathy.

## Figures and Tables

**Figure 1 diagnostics-11-00374-f001:**
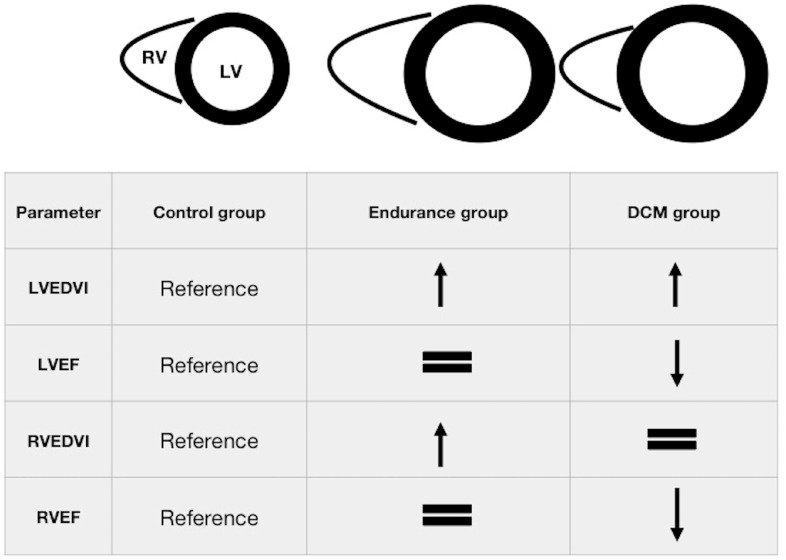
Graphical presentation of differences in the baseline CMR parameters between the studied groups. Please see Table 1 for the abbreviations.

**Figure 2 diagnostics-11-00374-f002:**
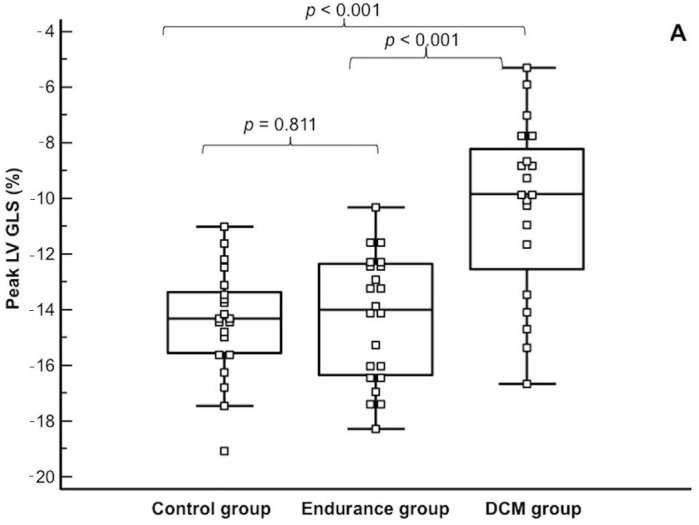
Comparison of the left ventricular peak global longitudinal (**A**), circumferential (**B**) and radial (**C**) strains between the studied groups. GCS—global circumferential strain, GLS—global longitudinal strain, GRS—global radial strain, LV—left ventricular.

**Figure 3 diagnostics-11-00374-f003:**
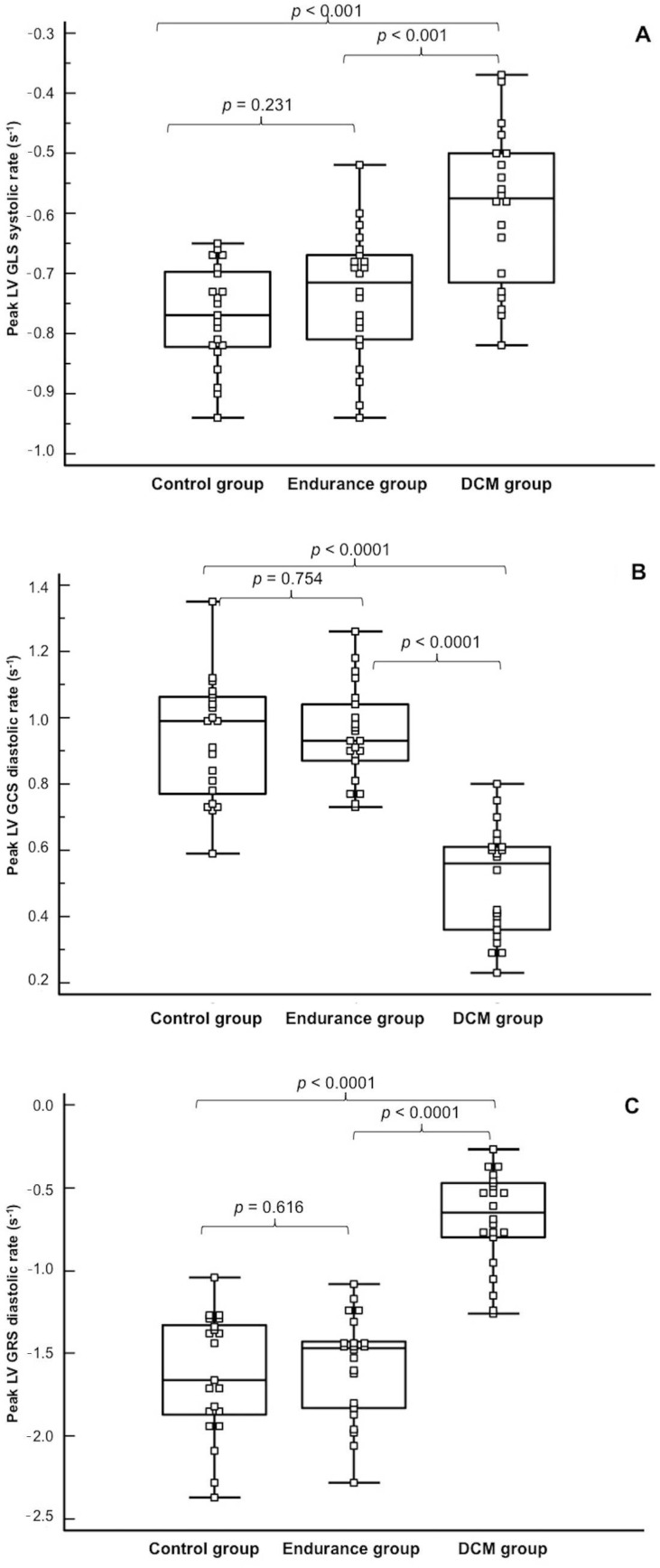
Comparison of the left ventricular peak global longitudinal systolic strain rate (**A**), global circumferential diastolic strain rate (**B**) and global radial diastolic strain rate (**C**) between the studied groups.

**Table 1 diagnostics-11-00374-t001:** Baseline cardiac magnetic resonance (CMR) parameters in studied groups.

Parameter	Control Group *n* = 21	Endurance Group *n* = 22	DCM Group *n* = 22	*p* for Trend
Age, years (±SD)	40 ± 12	41 ± 6	45 ± 12	0.271
LVEDVI, mL/m^2^ (±SD)	84 ± 15	117 ± 10 *	123 ± 29 *	<0.001
LVESVI, mL/m^2^ (±SD)	30 ± 6	43 ± 7	74 ± 14	<0.001
LVSVI, mL/m^2^ (±SD)	54 ± 10 #	74 ± 8	49 ± 13 #	<0.001
LVEF, % (±SD)	64 ± 4 **	63 ± 4 **	41 ± 9	<0.001
LVMI, g/m^2^ (±SD)	68 ± 10 #	86 ± 11	73 ± 14 #	<0.001
RVEDVI, mL/m^2^ (±SD)	89 ± 15 #	132 ± 18	88 ± 25 #	<0.001
RVESVI, mL/m^2^ (±SD)	39 ± 9 #	56 ± 11	44 ± 14 #	<0.001
RVSVI, mL/m^2^ (±SD)	50 ± 10 #	76 ± 10	44 ± 15 #	<0.001
RVEF, % (±SD)	56 ± 6 **	58 ± 5 **	49 ± 10	<0.001
LGE, % Nonischaemic Junction point Inferolateral IVS Ischaemic	1 (8) 1 1 0 0 0	4 (28) 4 3 1 0 0	11 (50) 11 1 4 6 0	0.003

* insignificant DCM vs. endurance, ** insignificant endurance vs. control, # insignificant DCM vs. control. DCM—dilated cardiomyopathy, IVS—interventricular septum, LGE—late gadolinium enhancement, LVEDVI—left ventricular end-diastolic volume index, LVEF—left ventricular ejection fraction, LVESVI—left ventricular end-systolic volume index, LVMI—left ventricular mass index, LVSVI—left ventricular stroke volume index, RVEF—right ventricular ejection fraction, RVEDVI—right ventricular end-diastolic volume index, RVESVI—right ventricular end-systolic volume index, RVSVI—right ventricular stroke volume index, SD—standard deviation.

**Table 2 diagnostics-11-00374-t002:** Deformation parameters of the left and right ventricles in the studied groups.

Parameter	Control Group *n* = 21	Endurance Group *n* = 22	DCM Group *n* = 22	*p* for Trend
**Left Ventricle**				
**Global longitudinal strain (±SD)** Peak, % Peak systolic rate, s^−1^ Peak diastolic rate, s^−1^	−14.4 ± 1.9 ** −0.77 ± 0.08 ** −0.78 ± 0.18	−14.3 ± 2.3 ** −0.73 ± 0.10 ** −0.56 ± 0.09 *	−10.3 ± 3.2 −0.59 ± 0.13 −0.57 ± 0.16 *	<0.001 <0.001 <0.001
**Global circumferential strain** Peak, % Peak systolic rate, s^−1^ Peak diastolic rate, s^−1^	−16.9 ± 2.0 −0.89 ± 0.14 −0.93 ± 0.18 **	−14.5 ± 2.6 −0.70 ± 0.12 −0.95 ± 0.14 **	−10.1 ± 2.7 −0.59 ± 0.14 −0.50 ± 0.16	<0.001 <0.001 <0.001
**Global radial strain (±SD)** Peak, % Peak systolic rate, s^−1^ Peak diastolic rate, s^−1^	27.6 ± 4.9 1.48 ± 0.42 −1.63 ± 0.37 **	22.6 ± 5.4 1.06 ± 0.30 −1.58 ± 0.31 **	14.6 ± 4.8 0.80 ± 0.24 −0.69 ± 0.29	<0.001 <0.015 <0.001
**Torsion, % (±SD)** Peak (°/cm) Peak rate (°/(cm·s))	0.86 ± 0.35 ** 7.7 ± 2.1 **	0.73 ± 0.36 */** 6.4 ± 2.1 */**	0.56 ± 0.19 * 5.6 ± 2.2 *	0.01 0.01
**Right Ventricle**				
**Global longitudinal strain (±SD)** Peak, % Peak systolic rate, s^−1^ Peak diastolic rate, s^−1^	−18.4 ± 4.1 −1.21 ± 0.31 1.03 ± 0.29	−15.7 ± 3.6 −1.00 ± 0.33 0.97 ± 0.30	−17.8 ± 5.5 −1.24 ± 0.74 1.08 ± 0.67	0.116 0.221 0.731

* insignificant DCM vs. endurance, ** insignificant endurance vs. control, # insignificant DCM vs. control.

**Table 3 diagnostics-11-00374-t003:** Receiver operator characteristic curve analysis of the CMR parameters to distinguish between physiology (athlete’s heart or controls) and pathology (dilated cardiomyopathy).

Variable	Cut-Off Value	AUC	Sensitivity	Specificity	*p*
RVEDVI	≤119 mL/m^2^	0.724	95.5%	41.9%	0.0004
LVEDVI	>122 mL/m^2^	0.732	50.0%	88.4%	0.0008
RVEF	≤50%	0.790	59.1%	93.0%	0.0001
Peak LV GLS systolic rate	>−0.65 s^−1^	0.830	70.0%	90.7%	0.0001
Peak LV GLS	>−11.0%	0.847	70.0%	97.7%	0.0001
Peak LV GRS	≤15.7%	0.918	72.7%	95.3%	0.0001
Peak LV GCS	>12.3%	0.931	81.8%	93.0%	0.0001
Peak LV GCS diastolic rate	≤0.70 s^−1^	0.974	90.9%	97.7%	0.0001
Peak LV GRS diastolic rate	>−1.27 s^−1^	0.987	100%	88.4%	0.0001
LVEF	≤53%	0.990	90.9%	100%	0.0001

AUC—area under the curve.

## Data Availability

The data presented in this study are available on request from the corresponding author.

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
