# Peer review of "Deformation Parameters of the Heart in Endurance Athletes and in Patients with Dilated Cardiomyopathy—A Cardiac Magnetic Resonance Study"

_diagnostics, 2021, doi:10.3390/diagnostics11020374_

Round 1

Reviewer 1 Report

The manuscript entitled “Deformation parameters of the heart in endurance athletes and in patients with dilated cardiomyopathy– a cardiac magnetic resonance study” aim to differentiate athlete’s heart and DCM by analyzing peak LV global longitudinal, circumferential, and radial strains (GLS, GCS, GRS respectively), peak LV torsion, and peak RV GLS using cardiovascular imaging. The manuscript is well written in general. Below are few comments that can further strengthen the manuscript:

  1. Page 9, line 261-263, looks like comments from the previous submission was included in the text. The authors should remove this and address the comments accordingly.
  2. GLS, GCS, GRS are the main methods used for analyzing the data. Representative figures (especially sections) of Figures 2 and 3 of each category would serve a great deal in the main text instead of just a video in the supplementary data. In addition, labeling for color gradient was missing in the video.

Author Response

Thank you very much for your review of our work. We have reviewed the manuscript and performed improvements (marked in red in text). Our answers to comments and the consequent changes are described in detail below.

1. Page 9, line 261-263, looks like comments from the previous submission was included in the text. The authors should remove this and address the comments accordingly.

We have accidentally left part of the template for submission provided by the publisher at the end of the discussion section explaining, what should be placed in this section. We have now removed it.

2. GLS, GCS, GRS are the main methods used for analyzing the data. Representative figures (especially sections) of Figures 2 and 3 of each category would serve a great deal in the main text instead of just a video in the supplementary data. In addition, labeling for color gradient was missing in the video.

Thank you for that comment. We have rephrased the sentence about the supplementary data in the results section: “Examples of main LV deformation parameters as 4D reconstructions in single subjects from each group are supplied in a supplementary file”. In fact, it is not a crucial figure, but just a set of examples of main LV deformation mechanics in 4D reconstruction in single subjects from each group. The software (cvi42, at least our version) does not provide color gradient for 4D models as demonstrated in the following source: https://www.lify.io/3p-products/cvi-42-ct-circle-cardiovascular-imaging. They can be found in other windows of the output format. We have used a trail version of the software, which has expired and we are unable to produce new movies with color gradient labelling. However, we have described the color labelling in the supplementary figure legend as follows: “The color gradient goes from high values (marked red) through yellow and green down to lowest values (marked blue)”.

Reviewer 2 Report

This study investigates the differences in LV function between healthy control, amateur endurance athletes, and age-matched patients with DCM. They found that the best value for differentiation between DCM and other groups was LV ejection fraction, peak LV GRS diastolic rate.

Differentiation between athlete’s heart and DCM has been reported in the past (Millar et al, Heart 2020;106:1059–1065, Galderisi et al, Eur Heart J Cardiovasc Imaging. 2015 Apr;16(4):353.)

They recommended exercise ECG and stress echocardiography. In this report, the authors focused more on Peak global circumferential and radial 96 strains (GCS, GRS) and peak GCS/GRS systolic and diastolic strain rates.

Although, the finding from the study is new in that it used rotational mechanics of hearts. I would suggest describing how rotational mechanics help differentiate these groups. Is it more useful for similar LVEF patients? How does it different from stress echocardiogram?

Here are my comments.

  1. In abstract, the authors should describe LVEF values and peak LV GRS values.
  2. In table 2, all the LV strain variables were significantly different. The authors may be able to conclude that LV strain variables were helpful to differentiate DCM subjects and other groups.
  3. If LVEF was specific enough to detect DCM, did you have to look into rotational mechanics of hearts?
  4. The past studies demonstrated that the stress test was helpful to differentiate these groups. In this study, there was clear difference in LVEF. Does this mean that the DCM patients were more advanced state than the previous studies?
  5. If the LVEF were similar, did rotational mechanics of hearts demonstrate significant difference?

Thank you for sharing the excellent study.

Author Response

Thank you very much for the review of our work. We have reviewed the manuscript and performed improvements (marked in red in text). Our answers to the comments and the consequent changes are described in detail below.

1. In abstract, the authors should describe LVEF values and peak LV GRS values.

We had to limit the number of words in the abstract to only 200 words as stated in the guide for authors. Therefore, in the abstract, we were forced to leave only the information on the best discriminatory cut-off values between controls/athletes and DCM for both LVEF and LV GRS.

2. In table 2, all the LV strain variables were significantly different. The authors may be able to conclude that LV strain variables were helpful to differentiate DCM subjects and other groups.

Thank you for that comment. We have changed the conclusion in the abstract and in the Conclusion section to: “There is a distinctive deformation pattern typical of each of the analysed groups, which could help in differentiating between athlete’s heart, non-athletic heart and dilated cardiomyopathy.“ We have also modified the first sentence of the Discussion section: “Our study is the first to analyse the differences in deformational mechanics of both ventricles of the heart in endurance athletes, non-athletic controls and patients with DCM.”

3. If LVEF was specific enough to detect DCM, did you have to look into rotational mechanics of hearts?

We wanted to identify rotational mechanics parameters that bear a potential value in discrimination between adaptive and maladaptive cardiac changes. In fact, we have demonstrated that despite marked differences in LVEF multivariable logistic regression model showed that the only independent predictor of DCM was peak LV GRS diastolic rate and not LVEF.

4. The past studies demonstrated that the stress test was helpful to differentiate these groups. In this study, there was a clear difference in LVEF. Does this mean that the DCM patients were more advanced state than the previous studies?

Yes, DCM patients in our study were in a more advanced state of the disease than in Millar et al study. However, the groups studied by Millar et al. might have been somewhat unusual for typical patients with DCM as commented also in a letter to the article by Stuart G et al. entitled in fact “Fascinating helpful article, but how typical were the patients with DCM and what does this tell us?” Some of those patients were exercising 8 hours per week with a mean of 4 hours, over 30% were off beta-blockers or ACEi/ARBs. Inclusion of patients with mild DCM morphology always brings a risk that a diagnosis might not be robust enough. Also as stated by Millar et al. in the response to this letter it is difficult of recruit individuals with DCM and mild morphology.

5. If the LVEF were similar, did rotational mechanics of hearts demonstrate significant difference?

Thank you for that comment. Given the findings from the multivariable analysis, we believe that at least LV GRS diastolic rate may be a potential additive discriminatory parameter between DCM and AH alongside LVEF. In fact, in our study LV GRS diastolic rate had 100% sensitivity and LVEF 100% specificity in discrimination between physiology and disease, showing their potential complementary value. However, this will have to be verified in a separate study on athletes and patients with milder forms od DCM. Following your comments we have added the sentence to the Discussion section: “Peak LV GRS diastolic rate and LVEF can be potentially considered as complimentary discriminatory parameters in differentiation between DCM and AH, as the first one had perfect sensitivity and the other one showed perfect specificity as demonstrated in our study”